# Decoupling Prompt Tuning for Long-Tailed Visual Recognition

## Abstract

Long-tailed visual recognition encounters difficulties in feature learning, particularly for tailed classes. Recently, fine-tuning visual prompt of pre-trained large models with powerful feature extraction capabilities for long-tailed data has been investigated. Although the result is promising, visual prompts are easily affected by semantically irrelevant parts of images, resulting in diminished effectiveness. To address this issue, we propose a Long-tailed Decoupling Prompt Tuning (LT-DPT) method for long-tailed visual recognition. LT-DPT explicitly decouples visual prompts into foreground-related prompts and background-related prompts, respectively. Specifically, foreground-related prompts emphasize saliency regions of the image which includes the most discriminative information for classification while background-related prompts capture common background features shared across classes, regardless of their category. In comparison with the state-of-the-art methods, extensive experiments demonstrate that the proposed method achieves better performance on long-tailed visual recognition benchmark datasets.

## 1 Introduction

In visual recognition, data sets may exhibit a long-tailed distribution (Van Horn et al., 2018; Yang et al., 2022), where a small number of classes (head classes) occupy most samples, while abundant classes (tail classes) contain only a few samples. Networks trained on such data typically exhibit poor generalization in tail classes (Zhang et al., 2023). To address this problem, a number of long-tailed visual recognition methods have been proposed in recent years, including data distribution re-balancing (Li et al., 2021; Park et al., 2022; Shi et al., 2023), loss adjustment-based methods (Kang et al., 2019; Li et al., 2022b; Tao et al., 2023), and representation improvement (Du et al., 2024; Jin et al., 2023). These approaches mitigate the challenges of long-tailed learning through diverse perspectives by improving training-from-scratch methods.

Instead of training-from-scratch methods, fine-tuning a pre-trained visual model, e.g., visual transformer (ViT) (Dosovitskiy et al., 2020), has been proposed to achieve efficient visual recognition (Chen et al., 2022; Hu et al., 2022). Along this line, visual prompt tuning (VPT) (Jia et al., 2022) has been exploited and shown to be promising for image classification, which freezes the pre-trained model and learns a small set of prompt parameters on the given data. Based on VPT, (Dong et al., 2023) introduced long-tailed prompt tuning (LPT), which learns class-shared and group-specific prompts to increase long-tailed learning performance. (Li et al., 2024) proposed a Gaussian neighborhood minimization prompt tuning method (GNM-PT) from an optimization perspective to improve the generalization of long-tailed learning.

Although these methods increase the ability of the pre-trained visual model for long-tailed visual recognition, the learned prompt is easily influenced by background information unrelated to the image content. For instance, we demonstrated the heatmaps obtained through Grad-CAM (Selvaraju et al., 2017) using the trained models by GNM-PT and our algorithm, respectively. As shown in Row 2 of Fig 1, prompt tuning in GNM-PT fails to focus on some areas with salient content. In contrast, attending to semantically significant parts (within red box in Row 3) can improve classification accuracy, e.g., $88.3\%$ for GNM-PT vs $89.0\%$ for Ours on CIFAR100-LT. Therefore, it is critical to guide visual prompts toward semantically relevant regions during fine-tuning, particularly for long-tailed visual recognition where discriminative cues are often subtle for some classes.

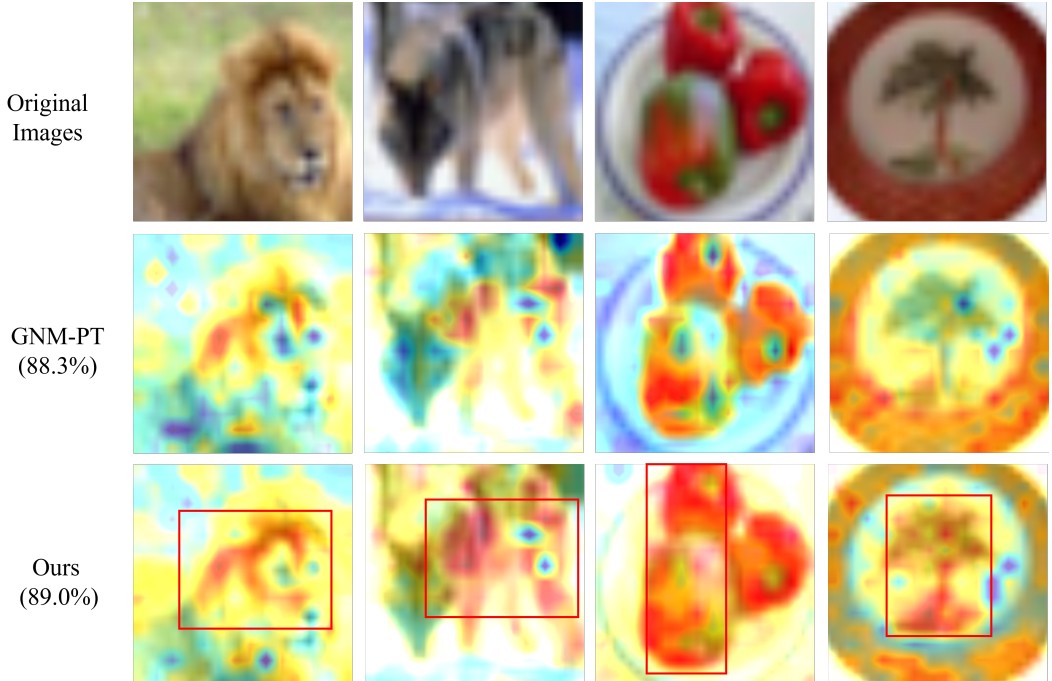

Figure 1: The heatmaps obtained through Grad-CAM using model trained by GNM-PT and our method, respectively. (a) The first row is original image; (b) The second row is the heatmaps obtained from GNM-PT; (c) The last row is the heatmaps obtained from ours. The red boxes show that our method focuses more on semantically related areas.

To address this issue, we propose a new visual prompt tuning method, namely long-tailed decoupling prompt tuning (LT-DPT), by decoupling visual prompts into foreground-related prompts and background-related prompts. Specifically, the foreground-related prompts are guided by saliency maps to attend to the most discriminative areas of the image, enabling the model to focus on features that are crucial for accurate classification. Conversely, we introduce a set of group prompts and construct the background-related prompts from them. The background-related prompts are encouraged to share context or common patterns present across different classes by the inverse saliency maps, which helps mitigate the negative impact of background bias and enhances the generalization ability of the model. Through the complementarity of two types of prompts, our method not only interacts with the shared background features among all classes, but also amplifies the influence of semantically meaningful features, thus reducing confusion between head and tail classes. Extensive experiments on benchmark datasets demonstrate that our method has great generalization ability on long-tail data, surpassing existing methods. The main contributions of this paper are summarized as follows: (1) we identify how self-attention in prompt tuning models is distributed across salient foreground and irrelevant background regions, uncovering its critical impact on long-tailed recognition; (2) we propose a method including foreground-related and background-related prompts, which use saliency-guided bias in self-attention to capture semantically related foreground and irrelevant background; (3) Comprehensive experiments demonstrate that our method outperforms its counterparts.

## 2 RELATED WORKS

### 2.1 LONG-TAILED VISUAL RECOGNITION

Extensive research has been conducted to address the negative effect of the long-tailed data distribution (Alshammari et al., 2022). The methods can be categorized into the following three directions: re-balancing methods, loss adjustment-based methods, and representation improvement methods. Data distribution re-balancing methods aim to mitigate class imbalance by balancing the training

data distribution, including specially designed balanced samplers (Cui et al., 2019; Kang et al., 2019) and data augmentation techniques (Wang et al., 2024; Shao et al., 2024), which are intuitive methods to fix the class gap with different sizes. However, these methods may cause overfitting or inject noisy samples, which therefore degrade overall model performance. Another popular technique is loss adjustment-based method, which improves the separability of different classes by re-weighting the loss function (Ren et al., 2020; Chen et al., 2023) or re-margining the margins for different classes (Wang et al., 2023; Zhao et al., 2022) according to their sample numbers. These methods adjust the output logits of the model to make it favor tail classes (Cao et al., 2019). Representation improvement methods (Li et al., 2022a;c) focus on increasing the quality of feature extraction, especially tail classes, to improve the generalization of the network. For example, balanced contrastive learning (Zhu et al., 2022; Du et al., 2024) improves model feature learning by comparing pairwise samples. Ensembling learning (Wang et al., 2021) aggregates diverse expert models to form complementary feature representations. Despite these advances, training-from-scratch method on long-tailed data remains challenging due to insufficient feature extraction and poor generalization for tail classes. Thus, leveraging pre-trained models and efficient fine-tuning strategies has become an active research direction.

## 2.2 EFFICIENT PROMPT TUNING

Efficient prompt tuning has emerged as a promising way to take advantage of the feature extraction power of large-scale pre-trained Transformer (Vaswani et al., 2017), as exemplified by ViT (Dosovitskiy et al., 2020) and CLIP (Radford et al., 2021). Instead of fine-tuning the entire model, pre-trained models achieve excellent performance in downstream tasks through fine-tuning a few trainable parameters (Houlsby et al., 2019; Zhou et al., 2022; Shi et al., 2024). In this paper, we focus on visual prompt tuning, which is a popular visual fine-tuning paradigm using ViT backbone with self-attention strategy (Lin et al., 2017). Inspired by prompt-based techniques in natural language processing, VPT first adapts visual prompts to the ViT pre-trained on ImageNet dataset (Deng et al., 2009) for image classification. By inserting learnable prompts into each layer, VPT allows the ViT backbone to extract task-relevant features while preserving its pre-trained knowledge. VPT has demonstrated strong performance with few trainable parameters and improved generalization for balanced visual recognition. Recognizing the challenges posed by long-tailed data distributions, LPT (Dong et al., 2023) explored visual prompt tuning in highly imbalanced training data and proposed a two-phase framework to improve long-tailed image classification performance. In LPT, a shared prompt learns general features and to adapt a pre-trained model into the target domain, while capture fine-grained differences among similar samples to improve discrimination ability. Different from LPT, GNM-PT (Li et al., 2024) designed a Gaussian neighborhood minimization optimizer for long-tailed visual prompt tuning from the perspective of the loss landscape, which improves model generalization while minimizing computational overhead.

## 3 PRELIMINARIES

### 3.1 VISUAL PROMPT TUNING.

VPT introduces $n_p$ prompt tokens $\boldsymbol{P} = [\boldsymbol{p}_1, \boldsymbol{p}_2, \ldots, \boldsymbol{p}_{n_p}] \in \mathbb{R}^{n_p \times D}$, where $D$ is the dimension of tokens in the pre-trained ViT. The learned prompt tokens encode the specific information in downstream tasks by collaborating with frozen patch representations obtained from ViT blocks. Depending on the number of Transformer layers involved, VPT has two versions: VPT-Shallow, which inserts prompt tokens at the first block only, and VPT-Deep, which inserts prompt tokens at every block. In this paper, we focus on VPT-Deep. The $l$-th layer $B$ of VPT-Deep is expressed as:

$$[\boldsymbol{z}_{cls}^l, \boldsymbol{Z}^l] = {\color{blue}B^l}([\boldsymbol{z}_{cls}^{l-1}, \boldsymbol{Z}^{l-1}, {\color{red}\boldsymbol{P}^{l-1}}]) \tag{1}$$

where $\boldsymbol{z}_{cls}$ is the class token and $\boldsymbol{Z} = [\boldsymbol{z}_1, \boldsymbol{z}_2, \ldots, \boldsymbol{z}_m]$ are the $m$ patch tokens, respectively. The colors ${\color{blue}blue}$ and ${\color{red}red}$ represent the frozen and learnable parameters, respectively.

In VPT, the prompts are able to extract task-relevant features from the patch tokens by self-attention mechanism. Ideally, the self-attention mechanism should guide prompts to aggregate information from the most discriminative patch tokens. We briefly introduce the self-attention mechanism below.

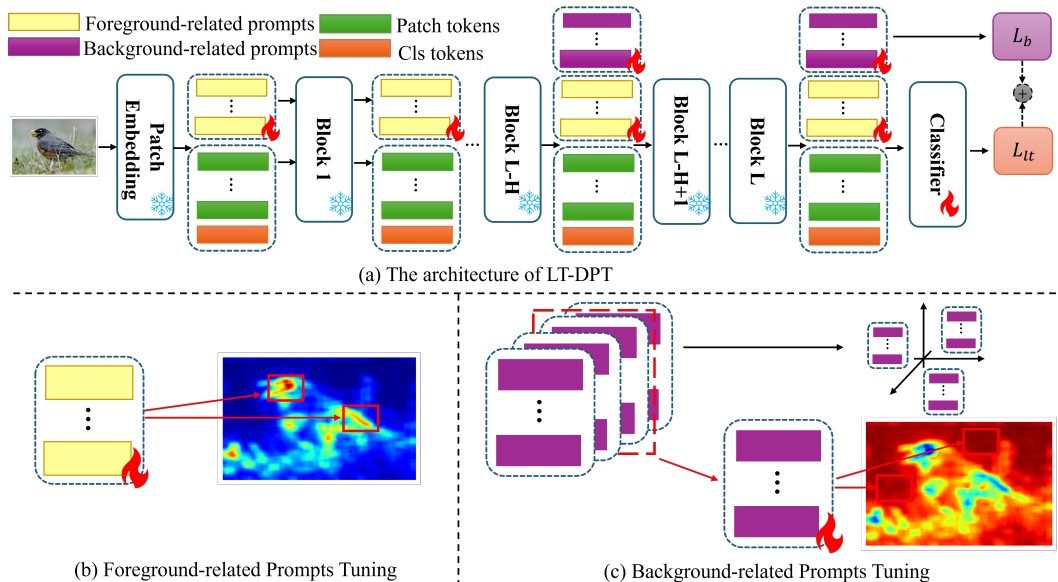

Figure 2: The framework of the proposed Long-tailed Decoupling Prompt Tuning (LT-DPT), where snow means frozen parameters and fire means trainable parameters. (a) LT-DPT consist of foreground-related prompts and background-related prompts. (b) Foreground-related prompts focus on the saliency regions; (c) Background-related prompts are selected from a group of prompts and emphasize the background regions.

## 3.2 SELF-ATTENTION MECHANISM.

In each layer of the model, the prompt token learns by interacting with the patch tokens through the self-attention mechanism. Specifically, each token is associated with a query, a key, and a value, which together facilitate information exchange. Self-attention can be described as mapping a query and a set of key-value pairs to an output. Given a prompt token $p_i$ with a corresponding query $Q_{p_i}$ and an arbitrary token $t$ characterized by a key $K_t$ and a value $V_t$, the attention output of $p_i$ is a weight sum of the value calculated as:

$$Attention(p_i) = Softmax(\frac{Q_{p_i}K_t}{\sqrt{D}})V_t, \tag{2}$$

As shown in equation 2, $p_i$ pays more attention to the tokens whose keys are more similar to its query. However, if the query of prompt token does not align closely with the keys corresponding to semantically significant or highly discriminative patch tokens, the resulting attention weights will not emphasize these regions. Consequently, the prompt token may fail to focus on the most informative or salient areas within the image, instead distributing its attention more uniformly or towards less relevant patches, thereby reducing model's discriminative ability.

## 4 PROPOSED METHOD

### 4.1 FRAMEWORK

The analysis in section 3.2 inspires us design a long-tailed visual prompt tuning method encouraging some prompt tokens focus on semantically significant patch tokens. Therefore, we propose a long-tailed decoupling prompt tuning method, which decouples the prompt tokens to two complementary parts. In our proposed method, one part of prompt tokens specializing to foreground regions and the other focusing more on background areas.

The framework of the proposed method LT-DPT is illustrated in Fig 2. For an $L$ layer model, we insert foreground-related prompts in all layers, while inserting background-related prompts in the last

$H$ layers. In $l$-th layer, we introduce a saliency-guided bias to query-key pairs for foreground-related prompts $\boldsymbol{P}_f^l = [\boldsymbol{p}_1^l, \boldsymbol{p}_2^l, \ldots, \boldsymbol{p}_{n_f}^l] \in \mathbb{R}^{n_f \times D}$ (Section 4.2). This bias makes the query of the prompt closer to the key of the significant patch token. For background-related prompts $\boldsymbol{P}_b^l = [\boldsymbol{p}_1^l, \boldsymbol{p}_2^l, \ldots, \boldsymbol{p}_{n_b}^l] \in \mathbb{R}^{n_b \times D}$, we design a group-specific prompts mechanism, where the patch embeddings of each image are associated with their corresponding group-specific prompts (Section 4.3). These group-specific prompts are specifically designed to capture the similar background information shared within each group, thereby enabling the model to more effectively exploit contextual cues from background regions.

## 4.2 FOREGROUND-RELATED PROMPTS

For foreground-related prompts in Fig 2, our objective is to guide these prompts to focus more effectively on salient regions within the patch tokens, thereby enhancing the prompts to extract discriminative features. Specifically, given the foreground-related prompts $\boldsymbol{P}_f^l$ with its query $\boldsymbol{Q}_{\boldsymbol{P}_f^l} \in \mathbb{R}^{n_f \times D}$, and the patch tokens $\boldsymbol{Z}^l = [z_1^l, z_2^l, \ldots, z_m^l]$ with key $\boldsymbol{K}_{\boldsymbol{Z}^l} \in \mathbb{R}^{m \times D}$ and value $\boldsymbol{V}_{\boldsymbol{Z}^l} \in \mathbb{R}^{m \times D}$, we add the normalized saliency map score (Hou & Zhang, 2007) $\boldsymbol{S}_{\boldsymbol{Z}^l}$ as bias in the query-key pairs, which is described as:

$$Attention(\boldsymbol{P}_f^l, \boldsymbol{K}_{\boldsymbol{Z}^l}, \boldsymbol{V}_{\boldsymbol{Z}^l}) = Softmax(\frac{\boldsymbol{Q}_{\boldsymbol{P}_f^l} \boldsymbol{K}_{\boldsymbol{Z}^l}}{\sqrt{D}} + \alpha \boldsymbol{S}_{\boldsymbol{Z}^l}) \boldsymbol{V}_{\boldsymbol{Z}^l}, \tag{3}$$

where $\alpha$ is a learnable scale parameter. Notably, saliency map score using (Hou & Zhang, 2007) provides a value for each pixel in the image, e.g., $32 \times 32$ map in CIFAR100-LT. To align the map with the number of patch tokens used in the backbone, we first divide the saliency map into a grid of non-overlapping patches, each of fixed size $\sqrt{m} \times \sqrt{m}$. Then, we use average pooling on each patch to get a $\sqrt{m} \times \sqrt{m}$ map. Finally, we normalize saliency map score $\boldsymbol{S}_{\boldsymbol{Z}^l}$ to lie within $[0, 1]$, ensuring that all bias values are non-negative. By explicitly introducing the normalized saliency information into the attention computation, we guide the foreground-related prompts to assign higher attention weights to salient patch tokens.

## 4.3 BACKGROUND-RELATED PROMPTS

For background-related prompts in Fig 2, we aim to facilitate the sharing of similar background information among these prompts. Inspired by the group-specific prompts proposed in LPT (Dong et al., 2023), we introduce $G$ prompt groups $\mathsf{P} = [\boldsymbol{P}_1^l, \boldsymbol{P}_2^l, \ldots, \boldsymbol{P}_g^l] \in \mathbb{R}^{G \times n_b \times D}$, where each group consists of $n_b$ learnable prompts with corresponding $n_b$ keys $\boldsymbol{K} = [\boldsymbol{k}_1^l, \boldsymbol{k}_2^l, \ldots, \boldsymbol{k}_{n_b}^l] \in \mathbb{R}^{n_b \times D}$ in the $l$-th layer. Given the embedding of the patch of an image $\boldsymbol{Z}$, the similarity $\boldsymbol{s} = [s_1, s_2, \ldots, s_{n_b}] \in [0, 1]^{1 \times n_b}$ to the prompts is calculated as the mean cosine similarity between each patch token and the keys:

$$Similarity(\boldsymbol{Z}, \boldsymbol{K}) = Mean(<\boldsymbol{Z}, \boldsymbol{K}>), \tag{4}$$

where $< \cdot, \cdot >$ is the cosine similarity operator. For the image $\boldsymbol{Z}$, we assume that the best matched $k$ similarities of the prompts are $\boldsymbol{s} = [s_1, s_2, \ldots, s_k]$, which correspond to $\mathsf{P} = [\boldsymbol{P}_{b1}^l, \boldsymbol{P}_{b2}^l, \ldots, \boldsymbol{P}_{bk}^l]$. Then, we set the background-related prompts as $\boldsymbol{P}_b^l = softmax(\boldsymbol{s})\mathsf{P}$, indicating that the background-related prompts are the weight sum of the most $k$ similar group prompts.

Similarly to the foreground-related prompts, we add normalized inverse saliency map score to the attention mechanism. For foreground-related prompts $\boldsymbol{P}_b^l$ with its query $\boldsymbol{Q}_{\boldsymbol{P}_b^l} \in \mathbb{R}^{n_b \times D}$ in the $l$-th layer, the attention score is for patch tokens $\boldsymbol{Z}^l$ is computed as:

$$Attention(\boldsymbol{P}_b^l, \boldsymbol{K}_{\boldsymbol{Z}^l}, \boldsymbol{V}_{\boldsymbol{Z}^l}) = Softmax(\frac{\boldsymbol{Q}_{\boldsymbol{P}_b^l} \boldsymbol{K}_{\boldsymbol{Z}^l}}{\sqrt{D}} + \beta(1 - \boldsymbol{S}_{\boldsymbol{Z}^l})) \boldsymbol{V}_{\boldsymbol{Z}^l}, \tag{5}$$

where $\beta$ is a learnable parameter. As shown in Eq 5, the background-related prompts are more oriented towards the background patch tokens guided by the normalized inverse saliency map.

To avoid all group prompts being similar and weaken the recognition ability of the model, we optimize each group prompt tend towards orthogonality. We design a group prompts orthogonal loss $L_b$, minimizing the cosine similarity between each pair of group prompts as:

$$L_b = \sum_{i \neq j}^{g} <\boldsymbol{P}_i, \boldsymbol{P}_j>. \tag{6}$$

Finally, we set the overall loss as the weight sum of long-tailed classification loss $L_{lt}$ and group prompts orthogonal loss $L_b$:

$$L = L_{lt} + \lambda L_b, \tag{7}$$

where $\lambda$ is a balance factor and will be discussed in the following.

## 5 EXPERIMENTS

### 5.1 DATASETS

**CIFAR100-LT** The original CIFAR-100 (Krizhevsky et al., 2009) is a computer vision dataset consisting of 60,000 images in 100 classes. The dataset includes 50,000 images for training and 10,000 for validation. In this paper, we follow the settings of Cao et al. (2019) to down-sample training samples per class while the validation dataset remains unchanged. The imbalanced ratio is calculated as the ratio between the numbers of samples in the most and least frequent classes. In our experiments, we set the imbalanced ratio as 200, 100, 50, and 10 for CIFAR-100-LT, respectively.

**iNaturalist2018** INaturailist2018 (Van Horn et al., 2018) is a large-scale fine-grained dataset in the real world that inherently exhibits a long-tailed distribution. The dataset consists of 437.5K training images in 8,142 classes with an imbalanced ratio of 512. The validation set contains 24.4K with 3 images per class. We follow the official splits of the dataset in our experiments.

**Places365-LT** Places365-LT (Liu et al., 2019) is a long-tailed version of the large-scale dataset Places365 (Zhou et al., 2017). Places365-LT consists of 62.5K training samples in 365 classes with an imbalanced ratio of 996 and 36.5K validation images.

**ImageNet-LT** Following (Liu et al., 2019), the long-tailed version of the large-scale dataset ImageNet (Russakovsky et al., 2015), called ImageNet-LT, is sampled from the original dataset using the Pareto distribution with a power parameter $\alpha$ set as 6. The ImageNet-LT contains 115.8K training images of 1,000 classes with an imbalanced ratio of 256. The validation set contains 50 images per class.

### 5.2 COMPARISON METHODS

We consider two types of state-of-the-art (SOTA) comparison methods: training-from-scratch methods and visual prompt tuning methods.

**Training-from-scratch methods.** We compare the SOTA training-from-scratch methods with the backbone ResNet (He et al., 2016). (1) Loss adjustment methods, i.e., LA (Menon et al., 2021), and GCL (Li et al., 2022b); (2) two-stage methods, i.e., LDAM-DRW (Cao et al., 2019), and Mis-LAS (Zhong et al., 2021); (3) Representation improvement methods, i.e., GPaCo (Cui et al., 2023), ProCo (Du et al., 2024), and PRL (Zhao et al., 2024b). Also, we consider the training-from-scratch method with the backbone ViT, i.e., LiVT (Xu et al., 2023).

**Visual prompt tuning methods.** We also compare the long-tailed visual prompt tuning methods with the backbone ViT, including the baseline VPT (Jia et al., 2022), the SOTA approaches LPT (Dong et al., 2023), and GNM-PT (Li et al., 2024).

### 5.3 IMPLEMENTATION DETAILS

Following (Dong et al., 2023), we use ImageNet-21K pre-trained ViT-B/16 (Dosovitskiy et al., 2020) and VPT-Deep for visual prompt tuning. We set GCL (Li et al., 2022b) as the classification loss function and use a cosine classifier. For the optimizer, we use SGD with GNM (Li et al., 2024) with a batch size of 128, a learning rate of 0.005, weight decay 0.01, and a momentum of 0.9, respectively. For our LT-DPT, the hyperparameter $H$ is set to 6 and the balance factor $\lambda$ is set to 0.5. Consistent with the long-tail visual prompt tuning, we utilize Deferred Re-Weighting (DRW) to achieve classifier re-balancing. For CIFAR100-LT, ImageNet-LT, iNaturalist2018, we fine-tune the model for 70 epochs, with the final 10 epochs for DRW. For Places365-LT, we use 100 epochs for fine-tuning and the last 40 epochs for DRW.

## 5.4 Experimental Results

We adopt Top-1 accuracy on test sets as the performance metric. We also provide accuracy measurements for three class splits based on the number of training data: Head ($> 100$ images), Medium (Med for short, $20{\sim}100$ images), and Tail ($\le 20$ images).

**Comparison on CIFAR100-LT.** We show the comparison results of Top-1 accuracy on CIFAR100-LT with different imbalance ratios in Table 1. For CIFAR100-LT, our proposed LT-DPT get significant improvements over previous methods, especially the training-from-scratch methods, across most cases of different imbalance ratios. We gain $+0.7\%$, $+0.3\%$, and $+0.5\%$ on CIFAR100-LT with imbalance ratios of 200, 100, and 50 compared to GNM-PT, respectively. Also, our LT-DPT also achieves good result when the imbalance ratio is 10, which gets the second-best among all compared algorithms.

Table 1: Comparison results on CIFAR100-LT with different imbalanced ratios in terms of Top-1 accuracy(%), where the best and second-best results are highlighted in **bold and underline**, and **bold**, respectively

| Imb Ratio | Backbone | 200 | 100 | 50 | 10 |
|-----------|----------|-----|-----|-----|-----|
| Training-from-scratch ||||||
| LA | ResNet | - | 43.9 | 47.0 | 57.7 |
| GCL | ResNet | 44.8 | 48.6 | 53.6 | - |
| LDAM-DRW | ResNet | 38.9 | 42.0 | 46.6 | 58.7 |
| MisLAS | ResNet | 43.5 | 47.0 | 52.3 | 63.2 |
| GPaCo | ResNet | - | 52.3 | 56.4 | 65.4 |
| ProCo | ResNet | - | 52.8 | 57.1 | 65.5 |
| PRL | ResNet | - | 52.8 | 57.3 | 65.6 |
| LiVT | ViT | - | 58.2 | - | 69.2 |
| Visual Prompt Tuning ||||||
| VPT | ViT | 72.8 | 81.0 | 84.8 | 89.6 |
| LPT | ViT | 87.9 | 89.1 | 90.0 | 91.0 |
| GNM-PT | ViT | **88.3** | **90.1** | **90.5** | **91.7** |
| **LT-DPT** | ViT | **89.0** | **90.4** | **91.0** | 91.6 |

Table 2: Comparison results on ImageNet-LT in terms of Top-1 accuracy(%), where the best and second-best results are highlighted in **bold and underline**, and **bold**, respectively

| Method | Head | Med | Tail | Overall |
|--------|------|-----|------|---------|
| Training-from-scratch |||||
| LA | 65.8 | 53.2 | 34.1 | 55.4 |
| GCL | 63.0 | 52.7 | 37.1 | 54.5 |
| LDAM-DRW | 60.4 | 46.9 | 30.7 | 49.8 |
| MisLAS | 62.9 | 50.7 | 34.3 | 52.7 |
| GPaCo | 67.4 | 57.1 | 41.2 | 58.9 |
| ProCo | 68.2 | 55.1 | 38.1 | 57.8 |
| PRL | - | - | - | 60.8 |
| LiVT | 73.6 | 56.4 | 41.0 | 60.9 |
| Visual Prompt Tuning |||||
| VPT | 79.5 | 76.5 | 72.8 | 77.2 |
| GNM-PT | **80.6** | **81.1** | **78.2** | **80.4** |
| **LT-DPT** | **85.8** | **84.1** | **78.4** | **84.0** |

**Comparison on ImageNet-LT.** We present the compared results of Top-1 accuracy on ImageNet-LT with three class splits in Table 2. As shown in Table 2, our proposed LT-DPT achieves superior performance, attaining an $84\%$ Top-1 classification accuracy, outperforming VPT and GNM-PT with a notable margin of $7.8\%$ and $3.6\%$, respectively. Notably, our method also achieves significant improvements across the head, medium, and tail classes, i.e., $+5.2\%$, $+3.0\%$, and $0.2\%$, respectively, demonstrating its outstanding performance.

**Comparison on iNaturalist2018.** Table 3 shows the performance of all the methods on iNaturalist2018. The proposed LT-DPT achieves a top-1 classification accuracy of $76.9\%$, surpassing both training-from-scratch methods and visual prompt tuning methods. Compared to visual prompt tuning methods, LD-DPT improves the performance of head, medium and tail classes, especially the medium classes.

**Comparison on Places365-LT.** From the Table 3, we can observe that LT-DPT shows an overall second-best performance on Places365-LT, is only inferior to GNM-PT. Compared to LPT, our

Table 3: Comparison results on iNaturalist2018 and Places365-LT in terms of Top-1 accuracy(%), where the best and second-best results are highlighted in **bold and underline**, and **bold**, respectively

| Method | iNaturalist2018 | | | | Places365-LT | | | |
|---|---|---|---|---|---|---|---|---|
| | Head | Med | Tail | Overall | Head | Med | Tail | Overall |
| Training-from-scratch | | | | | | | | |
| GCL | - | - | - | 72.0 | 38.6 | 42.6 | 38.4 | 40.3 |
| MisLAS | 73.2 | 72.4 | 70.4 | 71.6 | 39.6 | 43.3 | 36.1 | 40.4 |
| GPaCo | 73.0 | 75.5 | 75.7 | 75.4 | - | - | - | 41.7 |
| ProCo | **74.0** | 76.0 | 76.0 | 75.8 | - | - | - | - |
| PRL | - | - | - | 75.1 | - | - | - | 41.6 |
| LiVT | **78.9** | 76.5 | 74.8 | 76.1 | **48.1** | 40.6 | 27.5 | 40.8 |
| Visual Prompt Tuning | | | | | | | | |
| LPT | 62.1 | 76.2 | **79.3** | 76.1 | **47.6** | 52.1 | 48.4 | 49.7 |
| GNM-PT | 61.5 | **77.1** | **79.3** | **76.5** | 46.6 | **53.3** | **49.4** | **50.1** |
| **LT-DPT** | 62.7 | **77.7** | **79.5** | **76.9** | 46.9 | **52.6** | **49.2** | 49.9 |

method can balance the head, medium, and tail classes, improving the performance of the medium and tail classes.

## 5.5 ABLATION STUDY

**Effects of Different Components.** We consider the different impact of different components, i.e., general prompts without saliency score bias (G-prompts for short), foreground-related prompts (FR-prompts for short), and background-related prompts (BR-prompts for short). We also present the VPT as a baseline. We show the comparison performances of overall, head, medium, and tail classes on CIFAR100-LT with an imbalance ratio 200 in Table 4. In Table 4, method (A) is GNM-PT which differs from VPT in that it uses the GNM optimizer and long-tailed loss function. As observed in Table 4, foreground-related prompts, i.e., method (B), can effective improve the performance of head classes, leading an overall improvement of the results. In addition, it shows that the performances of the medium and tail classes get slight decrease. This is because foreground-related prompts focus more on the foreground, which leads to the model paying more attention to the dominant foreground of the head classes. By adding the background-related prompts, i.e., method (C), some common information can be shared within the samples, regardless of the classes. Therefore, all types of classes and the overall performance has been increased as shown in Table 4.

Table 4: Ablation studies of the different effects of each components on CIFAR100-LT with the imbalance ratio 200 in terms of Top-1 accuracy(%). The best and second-best results are highlighted in **bold and underline**, and **bold**, respectively.

| Method | G-prompts | FR-prompts | BR-prompts | Head | Med | Tail | Overall |
|---|---|---|---|---|---|---|---|
| VPT | ✓ | ✗ | ✗ | - | - | - | 72.8 |
| (A) | ✓ | ✗ | ✗ | 89.7 | **89.6** | **84.5** | 88.3 |
| (B) | ✓ | ✓ | ✗ | **90.4** | 89.5 | 84.5 | **88.4** |
| (C) | ✓ | ✓ | ✓ | **91.0** | **90.1** | **85.2** | **89.0** |

Table 5: Ablation studies of the different pre-trained model sizes on CIFAR100-LT with the imbalance ratio 200 in terms of Top-1 accuracy(%).

| Backbone | GNM-LT | | | | LT-DPT | | | |
|---|---|---|---|---|---|---|---|---|
| | Head | Med | Tail | Overall | Head | Med | Tail | Overall |
| ViT-T | 28.6 | 32.0 | 21.5 | 28.7 | 32.3 | 31.7 | 21.2 | 28.8 |
| ViT-S | 78.3 | 77.0 | 61.2 | 73.0 | 78.6 | 77.9 | 60.2 | 73.3 |
| ViT-B | 89.7 | 89.6 | 84.5 | 88.3 | 91.0 | 90.1 | 85.2 | 89.0 |

Table 6: Ablation studies of the different balance factor $\lambda$ in overall loss fuction on CIFAR100-LT with the imbalance ratio 200 in terms of Top-1 accuracy(%).

| $\lambda$ | 0 | 0.1 | 0.2 | 0.5 | 1.0 | 2.0 |
|---|---|---|---|---|---|---|
| Acc | 88.6 | 88.6 | 88.7 | 89.0 | 88.9 | 89.0 |

**Different Pre-trained Model Sizes.** We compare different pre-trained model sizes on CIFAR100-LT with the imbalance ratio 200 to verify the compatibility of our LT-DPT. Besides ViT-B/16 we used before, we utilize two smaller ImageNet-21K pre-trained ViT backbone, i.e., ViT-T/16 and ViT-S/16, respectively. We select the second-best algorithm, i.e., GNM-PT, on CIFAR100-LT as comparison method. As shown in Tab 5, all LT-DPT with different sizes exhibit overall better performance compared to GNM-LT, i.e., get a performance gain of $+0.1\%$, $+0.3\%$, and $+0.7\%$, respectively.

**Effects of Balance Factor $\lambda$.** The balance factor $\lambda$ is used to trade off between the long-tailed classification loss and group prompts learning during model training. To further investigate its effect, we examined model performance under different values of $\lambda$. Table 6 reports the Top-1 accuracy on CIFAR100-LT with an imbalance ratio of 200 under various settings of the balancing factor. As shown in Table 6, the performance of the balance factor on CIFAR100-LT is stable. A small balance factor can easily lead to redundancy in group-specific prompts, resulting in performance drop. The extreme case is $\lambda = 0$, meaning that no group-specific prompts constraint is employed. However, the performance is still better than the GNM-PT, i.e., $88.6\%$ vs $88.3\%$. This indicates that our complementary prompts are effective in long-tailed recognition.

## 6  CONCLUSION

In this paper, we have analyzed that the existing long-tailed prompt tuning methods tend to overlook semantically significant regions through self-attention mechanism. Based on this, we have proposed LT-DPT, which introduced two complementary prompts, i.e., foreground-related prompts and background-related prompts. Foreground-related prompts leverage normalized saliency map score as bias in query-key pairs to guide prompt focus on semantically related patch tokens. By contrast, background-related prompts are constructed by aggregating group-specific prompts with orthogonality constraint according to the similarity between patch tokens and learned prompt keys. With the guidance from a normalized inverse saliency map, the background-related prompts emphasize background regions. We have conducted extensive experiments and ablation studies to demonstrate the effectiveness of the proposed method and each component.

**Limitations and Future Works.** Although LT-DPT has been shown to be effective, it is not exempt from limitations. Recently, studies on visual prompt tuning with linguistic data have been proposed (Long et al., 2022; Zhao et al., 2024a). Therefore, how to use additional linguistic information to help guide prompt learning in our approach remains a challenge. Our future work will focus on the use of text information to guide prompt attention to important patch tokens in an effective way.

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

# A APPENDIX A. EFFECTS ON BACKGROUND-RELATED LAYER NUMBER $H$.

We compare the effects on different layers $H$ of background-related prompts in Tab 7. The Tab 7 shows that an excessively large number of layers, i.e., $H$=12, leads to degradation of model performance, meanwhile requiring a large tuning parameters. Also, it shows that an small number of layers, i.e., $H$=3, lead a poor performance.

Table 7: Ablation studies of the different effects of each components on CIFAR100-LT with the imbalance ratio 200 in terms of Top-1 accuracy(%). The best and second-best results are highlighted in **bold and underline**, and **bold**, respectively.

| $H$ | Tuned Params | Head | Med | Tail | Overall |
|---|---|---|---|---|---|
| 3 | **0.55M** | **90.7** | 89.3 | 83.1 | 88.1 |
| 12 | 1.94M | **90.7** | **89.9** | **85.0** | **88.9** |
| 6 | **1.01M** | 91.0 | 90.1 | 85.2 | 89.0 |

# B APPENDIX B. SALIENCY DETECTION

We introduce the saliency detection method used in this paper. Given a gray image $\boldsymbol{I}(x, y)$, the saliency map is calculated in the following step.

First, performing Fourier transform $\mathcal{F}(\cdot)$ on the image as:

$$\boldsymbol{M}(x,y)e^{j\boldsymbol{P}(x,y)} = \mathcal{F}(\boldsymbol{I}(x,y)), \tag{8}$$

where $\boldsymbol{M}(x,y)$ is the amplitude spectrum, and $\boldsymbol{P}(x,y)$ is the phase spectrum, respectively.

Then, the spectral residual $\boldsymbol{R}(x,y)$ is defined by:

$$\boldsymbol{L}(x,y) = log\boldsymbol{M}(x,y), \tag{9}$$

$$\boldsymbol{A}(x,y) = h(x,y) * \boldsymbol{L}(x,y), \tag{10}$$

$$\boldsymbol{R}(x,y) = \boldsymbol{L}(x,y) - \boldsymbol{A}(x,y), \tag{11}$$

where $h(x,y)$ is the filer, which we used a convolution filer in this paper.

Finally, the saliency map $\boldsymbol{S}(x,y)$ is computed as:

$$\boldsymbol{S}(x,y) = g(x,y) * \mathcal{F}^{-1}[exp(\boldsymbol{R}(x,y) + \boldsymbol{P}(x,y))]^2 \tag{12}$$

where $g(x,y)$ is a Gaussian filer, and $\mathcal{F}^{-1}$ is the Inverse Fourier Transform, respectively.

