# OpenReview forum: "Decoupling Prompt Tuning for Long-Tailed Visual Recognition"
_ICLR.cc/2026/Conference — ICLR 2026 Conference Withdrawn Submission_

### Official Review · Reviewer_Fw1V · 2025-10-28

**Soundness:** 3
**Presentation:** 4
**Contribution:** 3
**Rating:** 4
**Confidence:** 3

**Summary:**

This paper aims to address a key issue in long-tail visual recognition: existing Visual Prompt Tuning methods tend to be distracted by semantically irrelevant background information when learning features for tail classes (categories with few samples), which can negatively impact model performance.

**Strengths:**

[1] The idea is novel and intuitive: decoupling prompts into foreground and background components is both clever and straightforward. It directly addresses the shortcomings of existing methods, such as dispersed attention and susceptibility to background interference, providing a clear solution.
[2] The experimental validation is thorough: the authors conducted comprehensive experiments, comparing their method with various approaches—including training from scratch and prompt-tuning methods—across multiple datasets. They also designed detailed ablation studies to verify the effectiveness of foreground prompts, background prompts, and various hyperparameters (such as the weight λ of the orthogonal loss and the number of layers H where background prompts are applied), thereby strengthening the reliability of their conclusions.

**Weaknesses:**

[1] If the saliency detection algorithm used performs poorly on certain images—such as those where the subject blends with the background, the scene is complex, or the objects are very small—incorrect saliency maps may mislead the model and potentially degrade performance.
[2] The method simply divides image regions into foreground and background. However, in real-world images, elements such as the middle ground or context related to the main subject may also contain important discriminative information.
[3] It's unfair that the paper was trained on ImageNT21k but evaluated on ImageNet.

**Questions:**

[1] In scenarios where saliency detection is inherently challenging—such as with camouflage or harsh lighting—does the method’s performance degrade significantly?
[2] Could this “decoupling” concept be applied to other vision tasks? For example, in fine-grained image classification, could prompts for different components (such as a bird’s head or wings) also be decoupled? Would this approach be effective in object detection or segmentation tasks as well?
[3] What exactly are the “general patterns” learned by the background prompts, and to what extent do they help the model recognize tail classes?
[4] How does your method improve performance compared to LIFT?(Long-Tail Learning with Foundation Model: Heavy Fine-Tuning Hurts)

---

### Official Review · Reviewer_ss6g · 2025-11-01

**Soundness:** 2
**Presentation:** 3
**Contribution:** 1
**Rating:** 2
**Confidence:** 4

**Summary:**

This paper explores an interesting phenomenon: prompt tuning schemes tend to degrade model performance due to background attention bias under long-tail visual data distributions (characterized by imbalanced data categories) and proposes a method named Long-Tail Decoupled Prompt Tuning (LT-DPT). The core framework demonstrated by this method involves decoupling visual prompts into two complementary types: foreground-related prompts and background-related prompts. Foreground-related prompts are trained using the supervision of prior saliency maps, thus forcing the model to focus on the most discriminative object regions in images. Background-related prompts are guided to learn a shared generic background, maintaining diversity during actual inference through orthogonality loss. By concentrating on target regions and coordinating diversity and generalization in background prompts, the model achieves significant performance gains on classification tasks under long-tail distribution scenarios. The authors demonstrate the effectiveness of this decoupling strategy by surpassing existing state-of-the-art methods on multiple benchmark datasets.

**Strengths:**

The description of the design and approach in this paper is relatively clear.

The validation, parameter analysis, and ablation study are sufficiently comprehensive.

**Weaknesses:**

One of the core idea of this paper is to utilize saliency maps as supervisory signals, decoupling visual prompts into “foreground” and “background” components. While intuitive, this approach raises a significant concern: “Significant” and “usable for classification” do not exhibit a perfect correspondence, and the mask also contains some noise, which casts doubt on the validity of the supervision. From a logical standpoint, the performance ceiling of this method is likely to be significantly constrained by the quality of the significance detector. The paper does not appear to discuss the impact of the quality of the saliency map on the final performance.

The paper aims to provide a solution for long-tail visual recognition problems, but the solution proposed resembles a generic attention enhancement mechanism rather than a tailored solution targeting the core issue of class imbalance. The fundamental challenge in long-tail problem is the lack of diverse samples, which results in low-quality feature learning.
This approach enhances the model's focus salient regions within images, benefiting all categories(including head categories). However, it does not directly address the problem of learning discriminative representations for tail categories. Even if a model demonstrates strong attention guidance toward objects in a tail category, it still need to learn an ideal feature representation from an extremely limited number of samples. The method itself does not inherently achieve feature space rebalancing or enhancement of tail category representations, which are the key of solving the long-tail problem.
Essentially, the paper appears to be alleviating a symptom of the long-tail problem (e.g., distracted attention due to under-trained tail categories) rather than its root cause (insufficient representation learning stemming from data imbalance). The authors should provide a more in-depth analysis and establish a logical chain to demonstrate how separating foreground and background specifically alleviates the feature learning bottleneck for tail classes.

**Questions:**

Since the authors introduced external supervision signals based on saliency maps for prompt training, why not directly use the saliency maps as prompts, as demonstrated in the ECCV 2024 paper “Attention Prompting on Image for Large Vision-Language Models”? The author may need to provide relevant arguments to explain why their more complex prompt-based guidance mechanism outperforms these simpler, more direct masking strategies.

---

### Official Review · Reviewer_hP7a · 2025-11-03

**Soundness:** 2
**Presentation:** 3
**Contribution:** 2
**Rating:** 4
**Confidence:** 5

**Summary:**

This paper proposes a foreground–background decoupled visual prompt tuning method for long-tailed recognition. I thought the heuristic idea generally reasonable. However, the heuristic lacks solid justification whether theoretical or qualitative analysis. The only evidence of motivational is Fig. 1, which uses low-quality images and insufficient examples. In addition, recognition with saliency map has been studied (ref1). The manuscript does not offer insights beyond what is commonly known. It largely reads as a straightforward application of such ideas within visual prompt tuning. Given the modest compelling performance gains, I lean toward rejection.

[ref1] Decoupling Common and Unique Representations for Multimodal Self-supervised Learning, TPAMI2025

**Strengths:**

+ Problem framing is clear: the paper targets the foreground/background attention allocation issue in prompt tuning for long-tailed recognition and proposes saliency-guided attention biasing. The idea is intuitive and easy to follow.

**Weaknesses:**

- Heuristic nature and limited novelty. The core idea (i.e., decoupling prompts into foreground- and background-related components and injecting saliency/inverse-saliency as an attention bias) is essentially a heuristic perturbation to Q–K scoring in self-attention. The paper lacks principled justification, theoretical grounding, or new insights specific to long-tailed recognition or visual prompt learning.

**Questions:**

see Weaknesses

**Details Of Ethics Concerns:**

-

---

### Note · Authors · 2025-11-27

**Comment:**

We sincerely appreciate the reviewers' valuable comments. We will revise the manuscript to include more experiments and in-depth analyses to better explain how the proposed method addresses the bottleneck of long-tail prompt fine-tuning.

**Withdrawal Confirmation:**

I have read and agree with the venue's withdrawal policy on behalf of myself and my co-authors.